# Therapeutic Application of Brain-Specific Angiogenesis Inhibitor 1 for Cancer Therapy

**DOI:** 10.3390/cancers13143562

**Published:** 2021-07-16

**Authors:** Mitra Nair, Chelsea Bolyard, Tae Jin Lee, Balveen Kaur, Ji Young Yoo

**Affiliations:** 1Department of Neurosurgery, Mc Govern Medical School, The University of Texas Health Science Center at Houston, Houston, TX 77030, USA; Mitra.Nair@uth.tmc.edu (M.N.); Tae.Jin.Lee@uth.tmc.edu (T.J.L.); 2Department of Pediatric Surgery-Regenerative Medicine, McGovern Medical School, The University of Texas Health Science, Houston, TX 77030, USA; 3The Pelotonia Institute for Immuno-Oncology, The Ohio State University Comprehensive Cancer Center, The Ohio State University Wexner Medical Center, Columbus, OH 43210, USA; Chelsea.Bolyard@osumc.edu

**Keywords:** brain-specific angiogenesis inhibitor 1 (BAI1), oncolytic herpes simplex virus-1 (oHSV), glioblastoma (GBM), Vasculostatin (Vstat120)

## Abstract

**Simple Summary:**

Brain-specific angiogenesis inhibitor 1 (BAI1) is a transmembrane adhesion GPCR protein that plays an important role in many cellular processes and functions. The ability of BAI1 to promote anti-tumor and anti-angiogenic effects has been explored and developed as a treatment option for several different malignancies. Here, we have detailed a systemic overview of BAI, with a focus on its therapeutic potential for cancer. Due to the recent developments in oncolytic viruses and gene therapeutics towards targeting various types of cancers, our review article is highly relevant to clinical translation.

**Abstract:**

Brain-specific angiogenesis inhibitor 1 (BAI1/ADGRB1) is an adhesion G protein-coupled receptor that has been found to play key roles in phagocytosis, inflammation, synaptogenesis, the inhibition of angiogenesis, and myoblast fusion. As the name suggests, it is primarily expressed in the brain, with a high expression in the normal adult and developing brain. Additionally, its expression is reduced in brain cancers, such as glioblastoma (GBM) and peripheral cancers, suggesting that BAI1 is a tumor suppressor gene. Several investigators have demonstrated that the restoration of BAI1 expression in cancer cells results in reduced tumor growth and angiogenesis. Its expression has also been shown to be inversely correlated with tumor progression, neovascularization, and peri-tumoral brain edema. One method of restoring BAI1 expression is by using oncolytic virus (OV) therapy, a strategy which has been tested in various tumor models. Oncolytic herpes simplex viruses engineered to express the secreted fragment of BAI1, called Vasculostatin (Vstat120), have shown potent anti-tumor and anti-angiogenic effects in multiple tumor models. Combining Vstat120-expressing oHSVs with other chemotherapeutic agents has also shown to increase the overall anti-tumor efficacy in both in vitro and in vivo models. In the current review, we describe the structure and function of BAI1 and summarize its application in the context of cancer treatment.

## 1. Structure of Brain-Specific Angiogenesis Inhibitor 1 (BAI1)

Though BAI1 mRNA was originally reported only in the brain, it is now known that BAI1 expression is seen in a variety of tissue types in the body. Studies have shown its expression in the brain, pancreas, colon, stomach, kidney, and lung [1,2,3,4,5,6,7,8]. Decreased expression of BAI1 in these tissues has been correlated to increased tumor growth and vascularization. Though the exact relationship and mechanism between BAI1 and angiogenesis is unknown, one study showed that overexpression of BAI1 in vascular endothelial cells resulted in increased apoptosis [9], and the BAI-mediated inhibition of angiogenesis has been shown to occur following the interaction of its TSR domain with either CD36 scavenger receptors [10] or α_v_β_5_ integrin receptors [11] on endothelial cells. BAI1 is a transmembrane protein and a member of the adhesion G protein-coupled receptor (GPCR) family [12]. Like all adhesion GPCRs, BAI1 contains a large extracellular domain consisting of adhesive folds that allow for cell-to-matrix interactions, a GPCR autoproteolysis-inducing domain (also known as a GAIN domain), seven transmembrane regions, and an intracellular domain [13] (Figure 1).

Adhesion GPCRs are found on cell surfaces as non-covalently linked heterodimers. Upon proteolysis of the GAIN domain at its GPCR proteolysis site (GPS) [14], it creates an extracellular N-terminal fragment (NTF) and a membrane-spanning C-terminal fragment (CTF) which later re-associate in the cell membrane as heterodimers in the split GAIN domains [15,16,17]. It has been hypothesized that the reason adhesion GPCRs split into two fragments, only to re-integrate later, is partly to create two fragments that can behave independently from one another, and partly to create adhesion-GPCR heterodimers upon re-association [13,18].

The NTF of BAI1, also known as Vasculostatin (Vstat120) [19], comprises the original extracellular domain of the adhesion GPCR and a portion of the GAIN domain, while the CTF comprises the original intracellular domain of the protein, the transmembrane domain, and the other portion of the GAIN domain [13]. The NTF is a highly modular protein and plays a role at the interface of cell-to-cell and cell-to-matrix interactions [13,20]. The CTF interacts with G proteins and other protein partners and functions in intracellular signaling [13]. In addition to the basic features of adhesion GPCRs, BAI1 also contains a hormone-binding domain (HBD), five thrombospondin type 1 repeat (TSR) domains [11], and an integrin-binding Arg-Gly-Asp (RGD) motif on its NTF, as well as a Gln-Thr-Glu-Val (QTEV) motif on its CTF. These additional domains allow it to interact with other proteins involved in the modification of cytoskeletal architecture, as well as interact with the localization and trafficking of other proteins [21]. Interestingly, the CTF of BAI1 is rich in proline, which has been known to be involved with the regulation of signal transduction [22].

## 2. Functions of BAI1

### 2.1. Anti-Tumor and Anti-Angiogenic Activity

Angiogenesis is the process of new blood vessel formation and a known hallmark of cancer, providing a conduit for transporting nutrition and oxygen to cancer cells [23]. Since it is considered a pre-requisite for the growth of solid tumors, the inhibition of the pathways essential to driving angiogenesis has been harnessed for anti-cancer therapy [24]. The expression of BAI1 has been found to inversely correlate with tumor neovascularization and peritumoral brain edema [25]. This was initially identified during a screen for genes that are regulated by the tumor suppressor protein p53, and was implicated in the inhibition of neovascularization in glioblastoma [26]. However, while the BAI1 promotor regulatory sequence contains a p53 consensus binding site [27], the correlation between these two genes has not been substantiated in other studies [2,28]. Recently, Zhu et al. discovered that BAI1 prevents Mdm2-mediated p53 polyubiquitination. In this study, a loss of BAI1 expression through promoter methylation resulted in reduced p53 levels and increased tumor growth in medulloblastoma, a highly aggressive pediatric brain tumor [29]. It is important to note that epigenetic silencing due to the increased promoter methylation of BAI1 by the methyl-CpG-binding domain 2 (MBD2) has also been indicated in tumors; therefore, therapeutic strategies to silence MBD2 might lead to its reactivation [30]. As shown in Table 1, in addition to medulloblastoma and glioblastoma [2], BAI1 expression plays a significant role in other cancer types, including astrocytoma, renal cell carcinoma [28], pulmonary adenocarcinoma [1], colorectal cancer [6,7], bladder transitional cell carcinoma [27], metastatic brain cancer [31], and breast cancer [32]. BAI1 is frequently downregulated in these different cancer types and its level of expression has often been found to be inversely correlated with tumor malignancy. BAI1 expression is commonly seen in normal brain, colon, stomach, lung, pancreas, and kidney tissues and has higher expression in normal tissues compared to its malignant counterparts. Kudo, S et al. demonstrated the effects of BAI1 expression in mice inoculated with wild-type RCCs against mice inoculated with BAI1-expressing RCCs [4]. Overall, BAI1-expressing tumors showed significantly slower growth, decreased neovascular formation and decreased VEGF expression compared to wild-type tumors. Similarly, Duda, D.G. et al. examined the effects of induced BAI1 expression in pancreatic adenocarcinoma cells [3]. They illustrate the fact that BAI1-expressing tumor cells had lower expression of VEGF and MMP-1, which is known to be a positive regulator of tumor angiogenesis. Furthermore, they found that induced BAI1 expression in pancreatic adenocarcinoma cells exhibited slower tumor growth in vivo and also failed to establish a stable vascular network within the tumor compared to wild-type tumor cells. Liu et al. showed that decreased expression of BAI1 is correlated with poor prognosis in lung cancer, while overexpression of BAI1 inhibited tumor growth by inducing metabolic reprogramming via the SCD1-HMGCR module [33]. Collectively, these studies show that the reconstitution of BAI1 expression in tumors has the potential to be used as a therapeutic to reduce tumor growth and angiogenesis [19,34].

The above studies have all implicated BAI1 in modulating angiogenesis in human tumors. Studies using the panc-1 and renal cell carcinoma cell lines have also shown that overexpression of BAI1 suppresses tumor angiogenesis and tumor growth in animal models [3,4]. In one study, purified recombinant peptides encoding three of the five TSP-1 repeats in the NTF region of BAI1 were shown to inhibit rat corneal angiogenesis [26]. The ability for a transmembrane adhesion GPCR to function as an angiostatic factor, which are typically secreted, was resolved when a landmark study identified a conserved G protein proteolytic cleavage site (GPS) at the predicted stalk region of BAI1. This study identified Vstat120, a 120 kDa secreted extracellular fragment, within the conditioned medium of tumor cells engineered to overexpress BAI1 [19]. Corroborating these findings, a second study showed that overexpression of the extracellular fragment of BAI1 (BAI1-ECR) inhibited the proliferation of endothelial cells [11], and a third study showed that the lipid-mediated delivery of BAI1-ECR reduced corneal neovascularization [35]. Functionally, the inhibition of angiogenesis by BAI1 has been attributed to both the blockade of integrin signaling by its RGD motif and by its TSR domains present in the ECR [34]. Additionally, Vstat120 can be further reduced to a 40 kDa fragment (Vstat40) by matrix metalloproteinase-14 (MMP14) [36,37], producing a BAI1-derived angiogenesis inhibitor consisting solely of its RGD motif and the first of its five TSR domains [37]. There is also some evidence to suggest that MMP 14 can cleave BAI1 into Vstat 40 directly, without relying on the intermediary step of forming Vstat 120 [36,37]. Of the two, Vstat 40 is more commonly expressed in cell cultures and is thus the likely effector of BAI1′s anti-angiogenic activity [21]. The presence of TSR domains are of particular importance, as peptides with homologous domains have been shown to inhibit angiogenesis in vivo [38]. While angiogenesis is known to play an important role in tumorigenesis, and BAI1 expression is downregulated in certain cancer types [3], the precise mechanisms of BAI1-mediated anti-angiogenesis is still being elucidated. One study showed that overexpression of BAI1 in vascular endothelial cells resulted in increased apoptosis [9], and the BAI-mediated inhibition of angiogenesis has been shown to occur following the interaction of its TSR domain with either CD36 scavenger receptors [10] or α_v_β_5_ integrin receptors [11] on endothelial cells. As evidence for the latter scenario, BAI1 overexpression reduced the survival of human umbilical vein endothelial cells (HUVECs) by 51%. This reduction was rescued by anti-α_v_ integrin antibodies, but not anti-CD36 antibodies [11]. However, it has been shown that HUVECs have minimal expression of CD36, whereas HDMECs (human dermal microvascular endothelial cells) strongly express CD36 [39]. Furthermore, a recent study that examined the relationship between Vstat120 and CD36 indicated that Vstat120 inhibited the migration of HDMECs with little effect on HUVECs. This effect was rescued in the presence of neutralizing anti-CD36 antibodies [34]. Collectively, these provide evidence that the TSRs in BAI1 bind to both CD36 and α_v_β_5_ integrin receptors and inhibit angiogenesis, providing evidence that the BAI1 is an attractive therapeutic target for cancer therapy. Although BAI1 is known as an anti-angiogenic factor, the detailed mechanisms of its anti-tumor and anti-angiogenic activity have yet to be elucidated. More importantly, in clinical trials, most conventional anti-angiogenic drugs target the sprouting angiogenesis, and unfortunately these drugs have become resistant and have minimal clinical benefits [40]. Therefore, it would be interesting to study the mechanism of action of BAI1 depending on the type of angiogenesis, such as sprouting angiogenesis and intussusceptive angiogenesis.

### 2.2. Engulfment

In addition to angiogenesis, BAI1 plays an important role in the engulfment of apoptotic cells and infectious organisms by microglia [41]. Its presence on microglial cells acts as a receptor for phosphatidylserine [42], a membrane lipid that serves as an “eat me” signal on cells that should be engulfed by microglia [43]. BAI1 is integral to the formation of and ingestion by phagosomes during phagocytosis, and BAI1 knockdown microglia show a dramatic reduction in the engulfment of both apoptotic cells and bacteria [41]. While the TSP domains of BAI1 promote phagocytosis by binding to phosphatidylserine on the target cell, the C-terminus of BAI1 forms a trimeric complex with the proteins ELMO and Dock180 [44]. Specifically, the Dock180/ELMO complex acts as a guanine nucleotide exchange factor, resulting in the activation of the Rac GTPase protein [45], promoting a remodeling of the microglial actin cytoskeleton and ultimately enabling the engulfment of the target [44].

### 2.3. Myoblast Fusion

BAI1 also plays a role in myoblast fusion. A study by Hochreiter-Hufford et al. showed that the expression of BAI1 increased during myoblast fusion, and the overexpression of BAI1 enhanced the myoblast fusion via ELMO/Dock180/Rac1 signaling [46]. Interestingly, this study showed that BAI1 null mice were smaller in size and had significantly impaired muscle regeneration. However, ELMO and Rac were also shown to be critical for myoblast fusion, even when BAI1 was overexpressed, as knocking down ELMO2 protein expression in C2C12 myoblasts resulted in fewer and smaller myoblasts, and the inhibition of Rac via EHT 1864 also inhibited the myoblast fusion [46].

### 2.4. Synaptogenesis

BAI1 is also known to play a critical role in cell adhesion and signal transduction in the brain [47,48]. It has been shown to regulate cytoskeletal activity and signal transduction during neuronal growth [22] and regulate the release of neurotransmitters [47,48]. It also plays an important role in neural development, synapse formation, and signal transduction at neuronal synapses [49]. For example, BAI1 has been shown to regulate synaptogenesis by recruiting the Par3/Tiam1 polarity complex to synaptic sites, resulting in the activation of Rac1 GTPase, and ultimately regulating the development and plasticity of neuronal synapses and dendritic spines through the modulation of actin dynamics [50]. However, the mechanism by which BAI1 promotes synaptogenesis is completely distinct from those mechanisms by which it inhibits angiogenesis and promotes phagocytosis [50]. In the study by Zhu et al., they used BAI1 knockout (BAI1^−/−^) mice to show that BAI1 interacts with E3 ubiquitin ligase MDM2 and prevents the polyubiquitination and degradation of the postsynaptic density (PSD) component PSD-95, which regulates synaptic plasticity. Furthermore, they showed that BAI1^−/−^ mice have severe defects in their hippocampus-dependent spatial learning and memory and that the restoration of PSD-95 expression in hippocampal neurons in BAI1^−/−^ mice rescued defects in synaptic plasticity. Overall, this suggests a potential therapeutic application of BAI1 for neurological disorders [29].

## 3. Treatment Applications of BAI1

BAI1 expression inversely correlates with the malignancy and survival prognosis in several types of cancers (Table 1). Therefore, increasing BAI1 expression has been considered a possible method of cancer therapy, and has been specifically interrogated for the treatment of glioblastoma (GBM) [30], one of the most common and aggressive malignant brain tumors in adults [51]. Alteration in DNA methylation is known to increase tumor malignancy [52] and the DNA hypermethylation phenotype has been associated with GBM-harboring mutant IDH. Interestingly, BAI1 expression in GBM tissues has been shown to be regulated by methylation-mediated epigenetic silencing [30]. Following DNA methylation, the actual silencing of gene expression is accomplished by methyl-CpG-binding proteins, which bind to methylated DNA via the methyl-CpG-binding domain (MBD), enabling the transcriptional repression domain (TRD) to silence gene expression [53]. In a feedback loop, methylation of BAI1 in GBM results in the upregulation of MBD2 and its subsequent binding to BAI1′s methylated promoter, silencing BAI1 expression [30]. It has been reported that the treatment of glioma cell lines in vitro with the DNA demethylating agent 5-aza-2′-deoxycytidine (5-Aza-dC) restores BAI1 expression and reduces tumor growth [30]. In this same study, the inhibition of MBD2 using shRNA also restored BAI1 expression and reduced angiogenesis in an in vivo GBM xenograft model [30]. Collectively, these results indicate that both the restoration of BAI1 expression in tumors and targeting of MBD2 may be viable approaches for cancer treatment.

The direct administration of BAI1 is another approach that has been utilized successfully in the preclinical space (Table 2). As an example, Xiao et al. found that mice bearing GBM tumors and then treated with a recombinant adenovirus carrying the human BAI1 cDNA exhibited significantly increased survival times relative to control mice [54]. In another study, Kang et al. demonstrated the efficient transduction and successful therapeutic application of an adenoviral vector encoding BAI1 into established human GBM xenografts in SCID mice [51]. While the bulk of studies related to BAI1 are focused on its therapeutic effect against GBM, BAI1 has also been shown to have a therapeutic effect in other cancer types [3,4]. For example, Kudo et al. found that transfecting the BAI1 gene into a mouse renal cell carcinoma xenograft tumor model resulted in both decreased tumor growth and vascularity, compared to control mice [4]. In another study, Duda et al. found that the transfection of BAI1 reduced the growth and vascularity of pancreatic adenocarcinoma in vivo [3]. Taken together, these studies collectively indicate that BAI1 is a promising therapeutic approach for treating a variety of different cancer types, with a particular potential for success as a gene therapy agent for the treatment of GBM [51].

The direct transduction of Vstat120 has also been shown to result in anti-angiogenesis and tumor growth inhibition both in vitro and in vivo. For example, Kaur et al. found that when immunocompromised athymic nude mice were inoculated with human glioma cells expressing Vstat120, tumors were significantly smaller compared to tumors generated from unmodified glioma cells [30]. In another study, it was shown that Vstat120 can suppress intracranial tumor growth even in the presence of proangiogenic stimuli [34]. Similarly, it has been shown that loss of the tumor suppressor PTEN (phosphatase and tensin homolog) combined with expression of constitutively activating pro-tumorigenic mutations in EGFR (epidermal growth factor receptor) results in a highly aggressive and angiogenic glioma [58]. However, when these tumors are transfected with the expression vector for Vstat120, the survival times of the mice are indistinguishable from mice inoculated with less virulent glioma strains [34].

## 4. Application of Vstat120 in the Context of Oncolytic Virotherapy

Oncolytic virus (OV) therapy is of significant interest for treating patients with progressive or refractive GBM. Following the FDA approval of the oHSV talimogene laherparepvec (IMLYGIC^®^) for advanced melanoma patients, many different types of viruses have been developed and tested in preclinical and clinical studies, including those modified to express BAI1 or fragments thereof. For example, it has been shown that an adenovirus expressing BAI1 that is injected intratumorally into subcutaneous and intracerebral models of GBM resulted in extensive necrosis and reduced tumor vascularity [51]. A similar disruption of vascularity was seen with a BAI1-expressing adenovirus in a renal cancer model [4]. Additionally, our group has developed two different oHSVs armed with Vstat120 using two different generations of an oHSV. Both the first generation oHSV, RAMBO (rapid antiangiogenesis mediated by oncolytic virus), and the second generation oHSV, 34.5ENVE (viral ICP34.5 expressed by nestin promotor and Vstat120 expressing), were shown to have therapeutic efficacy in GBM models [59,60]. RAMBO showed effective oncolysis and decreased endothelial cell migration, tube formation, and microvessel density (MVD) both in vitro and in vivo. Evidence of secreted Vstat120 was found as early as 4 h post-infection in vitro and up to 13 days in vivo. Although RAMBO showed significant anti-tumor efficacy in mice bearing intracranial gliomas compared to mice treated with the control unarmed virus, only 20% had a complete response when used as a monotherapy. This was likely due to the attenuation of the first generation of the viral backbone [59,60]. The second-generation oHSV, 34.5ENVE, however, showed a significantly enhanced anti-angiogenic and anti-tumor efficacy compared to both RAMBO and an unarmed control virus. This is likely due to the fact that the second generation of the backbone allowed for an increased viral replication due to a tumor-specific transcriptional re-targeting [59,60]. As indicated in the previous sections, BAI1 is expressed on many different types of immune cells, such as macrophages and microglia, and plays a role in phagocytosis and bacterial pathogen clearance. Mechanistically, we have shown that BAI1 directs macrophage/microglia-mediated anti-viral responses, resulting in viral clearance, and limiting the therapeutic efficacy of oHSV. In this same study, however, we uncovered that Vstat120 expression from RAMBO virus shields viral particles from the BAI1-mediated inflammatory macrophage/microglia anti-viral response. This ultimately allows for an increased virus propagation and anti-tumor efficacy [61].

## 5. Utilization of Vstat120 in Combination with Bevacizumab in Glioblastoma

In addition, RAMBO has shown significant benefits as a therapeutic agent against GBM and other cancer types when used in combination with other anti-cancer agents. For example, Tomita et al. explored the use of RAMBO in combination with bevacizumab, an anti-vascular endothelial growth factor antibody (anti-VEGF), in a glioblastoma model [55]. GBM is among the most angiogenic of cancers, secreting high amounts of VEGF. When utilizing bevacizumab as a monotherapy against GBM, it proved to be ineffective due to the fact that a glioma invasion was induced through the activation of integrin pathways [62,63,64]. Tomita et al. demonstrated that bevacizumab increases the expression of cysteine-rich angiogenic inducer 61 (CCN1), an integrin-activating protein. The engagement of integrins by CCN1 leads to the activation of multiple downstream targets such as Akt, and ultimately results in tumor cell growth and invasion. However, upon combining bevacizumab with RAMBO, CCN1 expression was decreased. Tomita et al. showed that combining RAMBO with bevacizumab reduced Akt activation, illustrating the capacity for RAMBO to reduce the anti-VEGF-mediated activation of signaling pathways that encourage tumor growth and invasion. This study went on to show that the combination treatment of RAMBO plus bevacizumab was more effective than either therapy alone, both in vitro and in vivo. This was measured by increased tumor cell killing, increased anti-tumor efficacy, and decreased glioma cell invasion [55].

## 6. Utilization of Oncolytic Virus Expressing Vstat120 in Other Tumor Types

Another study examined the effect of copper chelation on the therapeutic efficacy of RAMBO in head and neck squamous cell carcinoma (HNSCC) [65]. Bis (2-hydroxyethyl) trimethylammonium (ATN 224) is a copper-chelating drug that induces cell death by disrupting metabolic activity in cells, specifically targeting the mitochondria [66]. Copper is known to be an important cofactor in the overall activation of angiogenesis and is found at higher levels in 40% of HNSCC patients. Interestingly, copper is also known to interfere with viral infection and replication. We have shown in previous studies that the presence of physiologic copper levels in culture has an inhibitory effect on RAMBO-mediated cell killing; however, when combined with ATN 224, the therapeutic effect of RAMBO was rescued both in vitro and in vivo [65].

The impact of doxorubicin on Vstat120-mediated oncolytic therapy was examined in mice bearing peritoneal ovarian cancer [56]. Doxorubicin is a chemotherapy drug which is used to treat a variety of cancers by acting as a DNA damaging agent. Combining doxorubicin with 34.5ENVE synergistically increased tumor cell killing through increased apoptosis in vitro and in vivo.

In addition to tumor cell killing, oHSV is thought to infect and kill proliferating tumor endothelial cells in vitro and in vivo [67]. Recently, using intravital in vivo imaging, we found that tumor-associated endothelial cells are able to effectively clear the virus without cell lysis, an effect that is rescued by the anti-angiogenic effect of RAMBO [57]. In addition, we demonstrated that RAMBO significantly inhibited oHSV therapy-induced endothelial cell activation both in vitro and in vivo, enhancing the anti-tumor efficacy and prolonging survival in heavily vascularized soft tissue sarcoma (STS)-bearing mouse xenografts [57]. In a separate study, we also found that RAMBO increases cisplatin retention in virus-infected ovarian cancer (OC) cells, inducing a DNA damage and inflammatory response, and sensitizing tumors to immune checkpoint inhibition [68].

## 7. Conclusions

BAI1 is a transmembrane adhesion GPCR protein that plays a role in many cellular processes and functions. The expression of BAI1 is frequently downregulated and is inversely correlated with tumor malignancy. The ability of BAI1 to promote anti-tumor and anti-angiogenic effects is one that has been previously explored and exploited for use as a treatment option against several different malignancies. This review provided an overview of the BAI1, with a focus on uncovered roles and the therapeutic potential of BAI1 for cancer therapy.

## Figures and Tables

**Figure 1 cancers-13-03562-f001:**
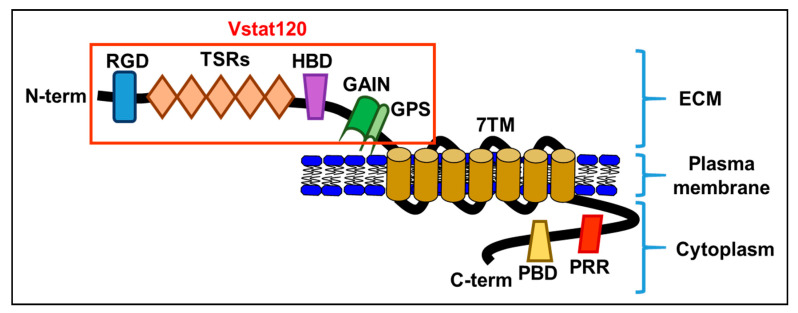
A schematic diagram of the brain-specific angiogenesis inhibitor (BAI) domains: RGD, integrin-binding Arg-Gly-Asp motif; TSRs, thrombospondin type 1 repeats; HBD, putative hormone-binding domain; GAIN, GPCR autoproteolysis-inducing domain; GPS, GPCR autoproteolysis site; 7TM, seven-transmembrane domain; PRR, proline-rich region; PBD, PDZ-binding domain. Vstat120, secreted extracellular BAI1.

**Table 1 cancers-13-03562-t001:** The list of references shows that BAI1 expression is inversely correlated with the tumor malignancy and survival prognosis in cancers.

References	Cancer Type	Sample	*n*	Results	Correlation with Disease?
[1]	Pulmonary Adenocarcinoma	Human Pulmonary Adenocarcinoma Tissue Samples	48	BAI1 protein expression found in 38/48 samples, with vascularity inversely proportional to BAI1 express	BAI expression is inversely correlated with
[2]	Glioblastoma	Human Glioblastoma Cell Lines	37	BAI1 protein expression in 13/37 of cell lines	BAI1 expression is inversely correlated
[5]	Gastric Cancer	Human Gastric Cancer Tissue Samples	200	No significant difference in BAI1 expression in either normal tissues or cancers	BAI1 expression is not correlated
[6]	Colorectal Cancer	Tissues from colorectal cancers and extraneoplastic colon mucosa	102 (62 colorectal cancers and 40 extraneoplastic colon mucosa)	BAI1 protein expression was slightly decreased in the cancer tissue, suggesting specific types of angiogenic factors have a protective roles against cancer	BAI expression is inversely correlated
[7]	Colorectal Cancer	Human Colorectal Cancer Cell Lines	49	BAI1 protein expression is significantly reduced, and its expression is inversely correlated to vascular invasion and metastasis	BAI1 expression is inversely correlated
[8]	Gastric Cancer	Human Gastric Cancer Tissue Samples	32	BAI1 protein expression decreased in cancer tissue and metastatic lymph node tissue relative to extraneoplastic mucosa and non-metastatic lymph node tissue	BAI1 expression inversely correlated
[25]	Astrocytoma	Human Tissue Samples	101 (90 human brain astrocytoma specimens, 11 normal human brain tissue specimens)	BAI1 protein expression decreased as tumor grade increased	BAI1 expression is inversely correlated
[27]	bladder transitional cell carcinoma (BTCC)	human BTCC biopsy specimens	131	BAI1 protein expression is negatively correlated with BTCC angiogenesis, and its expression is associated with reduced p53 mutations.	BAI expression is inversely correlated
[28]	Renal Cell Carcinoma	Human Tissue Samples	57 (32 localized carcinomas, 15 advanced carcinomas, 10 normal kidney tissue specimens)	BAI1 protein expression in 31/32 localized carcinomas, and 9/15 in advanced carcinomas	BAI1 expression is inversely correlated
[30]	Glioblastoma	Human Glioblostoma Tissue Samples	424	Consistent and dramatic reduction in the expression of BAI1	BAI expression is inversely correlated
[31]	brain metastasis	Tissues from brain metastasis of primary adenocarcinoma of the lung	2	Decreased BAI1 expression is involved in the periphery-to-brain metastasis	BAI expression is inversely correlated
[33]	Lung Cancer	Human Lung Cancer Cell Lines	103 (primary lung tumor tissues)	BAI1 functions as a tumor suppressor by inducing metabolic reprogramming in lung cancer	BAI expression is inversely correlated

**Table 2 cancers-13-03562-t002:** The list of references shows the application and its therapeutic efficacy of BAI1.

Model	Therapeutic	Experimental Read Out	Results/Outcome	References
Human pancreatic adenocarcinoma cell lines in vitro and in vivo	Transfection of BAI1 gene into pancreatic adenocarcinoma cell lines by means of adenoviruses	Tumor growth	No difference in tumor growth between treated and untreated cells in vitro Diminished tumor growth (*p* < 0.05)	[3]
Mouse Renal Cell Carcinoma (Renca) cell line xenografted into BALB/c mice	Transfection of Renca cells with BAI1	Tumor Size and Tumor Blood Flow	Decreased tumor size and blood flow (*p* < 0.01)	[4]
Human glioblastoma cells xenografted into mice	Induced expression of vasculostatin in glioblastoma cells	Vascular Channel Length	Vascular channel length is decreased (*p* < 0.03)	[19]
Human Glioblastoma Cells	Transfection of glioma cells with BAI1	Tumor cell migration	Decreased tumor cell migration (*p* < 0.001)	[30]
Human Lung Cancer	Stably BAI1 overexpression in lung cancer cells	Tumor cell migration, colony formation, tumor cell growth in vitro and in vivo	Decreased tumor cell migration, colony formation, and tumor growth (*p* < 0.001)	[33]
Human intracranial glioma cells xenografted into mice	Transfection of glioma cell lines with Vasculostatin 120 expression vectors	Survival in mice and Vascular density of gliomas	Increased survival time (*p* < 0.05) Decreased blood vessel density (*p* < 0.05)	[34]
Corneal neovascularization in an in-vivo rabbit model	subconjunctival injection of the BAI1-ECR gene mixed with nonliposomal lipid	Neovascularized area	Less neovascularized area (*p* < 0.05)	[35]
Human glioblastoma cells xenografted into SCID mice	In vivo transduction of transplanted tumors by adenoviral vector encoding BAI1 (AdBAI1)	Neovascularization post transplant and Tumor Growth	Angiogenesis completely inhibited and Tumor growth inhibited (*p* < 0.05)	[51]
Human glioblastoma cells xenografted into in-vivo mouse models	Recombinant adenovirus carrying human BAI1 cDNA	Survival in mice	Increased survival (*p* < 0.05)	[54]
Human intracranial glioma cells implanted into athymic nude mice	Combination treatment of Vasculostatin expressing virus and Bevacizumab	Survival in mice	Increased survival time (*p* < 0.05)	[55]
Human ovarian cancer cells implanted into athymic nude mice	Combination treatment of Vasculostatin expressing virus and Doxorubicin	Survival in mice	Increased survival time (*p* < 0.001)	[56]
Human soft tissue sarcoma and glioblastoma cells implanted into athymic nude mice	Vstat120-expressing RAMBO virus decreases tumor endothelial cell activation, inreasing anti-tumor efficacy	Tumor growth	Increased anti-tumor efficacy (*p* < 0.001)	[57]

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
