# Peer review of "Therapeutic Application of Brain-Specific Angiogenesis Inhibitor 1 for Cancer Therapy"

_cancers, 2021, doi:10.3390/cancers13143562_

Round 1

Reviewer 1 Report

The article needs careful review:

1 - after describing the structure of BAI1 – it would be correct to indicate the distribution on tissues/organs of BAI1, and this must be correlated with the location of tumors in which BAI1 is deficient;

2 - table 1 - the last two articles are identical;

3 - the articles mentioned in the paragraph preceding table 1 - are not found in their entirety in the table [ex - citations 23, 17, 25 are not in table];

4 - citations are not always correct - ex: the cited article number 33 is not correct:

The text from 33:

The N terminus of BAI1 has also been shown to undergo further processing by a furin / matrix metalloproteinase-14 protease cascade to release a 40-kDa fragment, vasculostatin 40, which also inhibits angiogenesis (18).

it is in fact the cited article no. 18 from the cited article 33.

  1. Stephenson, J.R., Paavola, K., Schaefer, S.A., Kaur, B., Meir, E.G., & Hall, R.A. (2013). Brain-specific Angiogenesis Inhibitor-1 Signaling, Regulation, and Enrichment in the Postsynaptic Density*. The Journal of Biological Chemistry, 288, 22248 - 22256.

18 FROM 33. Cork, S. M., Kaur, B., Devi, N. S., Cooper, L., Saltz, J. H., Sandberg, E. M., Kaluz, S., and Van Meir, E. G. (2012) A proprotein convertase/MMP-14 proteolytic cascade releases a novel 40 kDa vasculostatin from tumor suppressor BAI1. Oncogene 31, 5144–5152

5 - the description of MYOBLAST FUSION - refers to an article by Hochreiter - Hufford - which is not found in the bibliography - instead the article from position 43 is quoted - which is something else entirely;

Hochreiter-Hufford, A. E., Lee, C. S., Kinchen, J. M., Sokolowski, J. D., Arandjelovic, S., Call, J. A., Klibanov, A. L., Yan, Z., Mandell, J. W., & Ravichandran, K. S. (2013). Phosphatidylserine receptor BAI1 and apoptotic cells as new promoters of myoblast fusion. Nature, 497 (7448), 263–267. https://doi.org/10.1038/nature12135

6 - table 2 - the mentioned articles are not found in the explanatory text, and from the explanatory text, not all the citations are found in the table [eg - 52, 53, 21 - are not found in the table];

7 - it would be correct to indicate in the tables the cited articles from the references list;

8 - the last article in table 2 is not found in the references list.

Author Response

  1. After describing the structure of BAI1 – it would be correct to indicate the distribution on tissues/organs of BAI1, and this must be correlated with the location of tumors in which BAI1 is deficient;

Response: We thank the reviewer for the suggestion. In the revised manuscript, we have added the distribution of BAI1 as below.

Though BAI1 mRNA was originally reported only in the brain, it is now known that BAI1 expression is seen in a variety of tissue types in the body. Studies have shown expression in brain, pancreas, colon, stomach, kidney and lung (1-7). Decreased expression of BAI1 in these tissues have been correlated to increased tumor growth and vascularization. Though the exact relationship and mechanism between BAI1 and angiogenesis is unknown, one study showed that overexpression of BAI1 in vascular endothelial cells resulted in increased apoptosis (8), and BAI-mediated inhibition of angiogenesis has been shown to occur following the interaction of its TSR domain with either CD36 scavenger receptors (9) or αvβ5 integrin receptors (10) on endothelial cells.

  1. Table 1- the last two articles are identical;

Response: We apologize for this mistake. We have corrected this mistake in the revised manuscript.

  1. The articles mentioned in the paragraph preceding table 1 - are not found in their entirety in the table [ex - citations 23, 17, 25 are not in table];

Response: We apologize for this mistake. We have corrected this mistake and added 3 references in the table 1 in the revised manuscript.

  1. Citations are not always correct - ex: the cited article number 33 is not correct:

The N terminus of BAI1 has also been shown to undergo further processing by a furin / matrix metalloproteinase-14 protease cascade to release a 40-kDa fragment, vasculostatin 40, which also inhibits angiogenesis (18).

it is in fact the cited article no. 18 from the cited article 33.

Response: We apologize for this mistake. We have corrected this mistake in the revised manuscript.

  1. the description of MYOBLAST FUSION - refers to an article by Hochreiter - Hufford - which is not found in the bibliography - instead the article from position 43 is quoted - which is something else entirely;

Hochreiter-Hufford, A. E., Lee, C. S., Kinchen, J. M., Sokolowski, J. D., Arandjelovic, S., Call, J. A., Klibanov, A. L., Yan, Z., Mandell, J. W., & Ravichandran, K. S. (2013). Phosphatidylserine receptor BAI1 and apoptotic cells as new promoters of myoblast fusion. Nature, 497 (7448), 263–267. https://doi.org/10.1038/nature12135

Response: We apologize for this mistake. We have corrected this mistake in the revised manuscript.

  1. Table 2 - the mentioned articles are not found in the explanatory text, and from the explanatory text, not all the citations are found in the table [eg - 52, 53, 21 - are not found in the table];

Response: We apologize for this mistake. In the revised manuscript, we added reference #21 to the table and deleted references # 52 and 53 from the article.

  1. It would be correct to indicate in the tables the cited articles from the references list;

Response: We thank the reviewer for the suggestion. In the revised manuscript, we have added reference numbers in the table.

  1. The last article in table 2 is not found in the references list.

Response: It was listed in the reference # 64 and # 65. In the revised version, those are listed in the reference # 63 and #65. Thank you.

Reviewer 2 Report

In the light of research on the role of angiogenesis in the cancer pathogenesis , specific angiogenesis inhibitors may have potential use in anticancer therapy.

The layout of the manuscript is typical for review, and the contents of the subsections consistently reflect the purpose of this work.

Prepared tables collect information about the expression and therapeutic efficiency of BAI1 in a clear and reliable way. The scheme very vividly shows structure of brain-specific Angiogenesis Inhibitor 1.

References include 67 items, publications have been selected and cited correctly. Articles from recent years are included.

Author Response

Thank you.

Reviewer 3 Report

The review by Nari et al aims at giving an overview of the role of BAI1 as a potential actor for designing new cancer treatment. By only reading the manuscript, I understand that the authors have specific expertise on this field. This could be the reason why the manuscript lacks from important logical connections and information. However, a well-written and organized review should give to the reader all the relevant tools for obtain the most for him/her research/clinical practice. More in particular, some important points are:

  • The authors often write that BAI1 is involved in important processes, such as angiogenesis. It could be important to give to the readers some principal information about how BAI1 acts in these mechanisms (for example in the Introduction???)
  • Section 2.1: the authors did not discuss experimental evidence regard the role of BAI1 as anti-tumor/angiogenic element. Furthermore, the authors report only a list of histotypes with the relative references.... I have some doubts about that this paragraph could improve the quality of knowledge of the readers
  • Sections 2.2, 2.3 and 2.4 should be more focused on cancer
  • Section 3: as the authors said, the most relevant data on BAI1 and cancer treatment are on GBM. I suggest the authors to separate a section of evidence on GBM and another one on the other cancer types 

My minor comments are:

  • English should be extensively received as well as typing and styling errors
  • Please remove the title of the reference, and replace with the number of the reference, in order to render both the tables are readable 
  • References should appear at the end of the sentences

Author Response

Reviewer #3: The review by Nair et al aims at giving an overview of the role of BAI1 as a potential actor for designing new cancer treatment. By only reading the manuscript, I understand that the authors have specific expertise on this field. This could be the reason why the manuscript lacks from important logical connections and information. However, a well-written and organized review should give to the reader all the relevant tools for obtain the most for him/her research/clinical practice. More in particular, some important points are:

  1. The authors often write that BAI1 is involved in important processes, such as angiogenesis. It could be important to give to the readers some principal information about how BAI1 acts in these mechanisms (for example in the Introduction???)

Response: We thank the reviewer for the suggestion. We have added the action mechanism of BAI in the introduction section in the revised manuscript as below.

Decreased expression of BAI1 in these tissues have been correlated to increased tumor growth and vascularization. Though the exact relationship and mechanism between BAI1 and angiogenesis is unknown, one study showed that overexpression of BAI1 in vascular endothelial cells resulted in increased apoptosis (8), and BAI-mediated inhibition of angiogenesis has been shown to occur following the interaction of its TSR domain with either CD36 scavenger receptors (9) or αvβ5 integrin receptors (10) on endothelial cells.

  1. Section 2.1: the authors did not discuss experimental evidence regard the role of BAI1 as anti-tumor/angiogenic element. Furthermore, the authors report only a list of histotypes with the relative references.... I have some doubts about that this paragraph could improve the quality of knowledge of the readers.

Response: We apologize not providing the details. We have updated this section.

  1. English should be extensively received as well as typing and styling errors.

Response: We have corrected this typing error and also revised the entire manuscript thoroughly for any typo and grammatical errors.

  1. Sections 2.2, 2.3 and 2.4 should be more focused on cancer

Response: We thank the reviewer for the suggestion. In the revised manuscript, we have corrected.

  1. Section 3: as the authors said, the most relevant data on BAI1 and cancer treatment are on GBM. I suggest the authors to separate a section of evidence on GBM and another one on the other cancer types 

Response: We thank the reviewer for the suggestion. In the revised manuscript, we separated sections into GBM and other cancer types.

My minor comments are:

  1. English should be extensively received as well as typing and styling errors.

Response: We have corrected this typing error and also revised the entire manuscript thoroughly for any typo and grammatical errors.

  1. Please remove the title of the reference, and replace with the number of the reference, in order to render both the tables are readable 

Response: We thank the reviewer for the suggestion. In the revised manuscript, we have changed tables as suggested and replaced with new format of tables.

  1. References should appear at the end of the sentences.

Response: We apologize for this mistake. We have corrected this in the revised manuscript.

Reviewer 4 Report

In the manuscript entitled “Therapeutic Application of Brain-Specific Angiogenesis Inhibitor 1 for Cancer Therapy” by Mitra Nair, Chelsea Bolyard, Tae Jin Lee, Balveen Kaur *, Ji Young Yoo*, the authors review structure and function of the Brain-specific angiogenesis inhibitor 1 (BAI1/ADGRB1), an adhesion G protein-coupled receptor, and its application in the context of cancer therapy. They present the structure and functions of BAI1/ADGRB1 and summarize the works demonstrating that BAI1/ADGRB1 expression is inversely correlated with the tumour malignancy and survival prognosis in cancer. Likewise, they take into account other roles of BAI1/ADGRB1 on phagocytosis, myoblast fusion and synaptogenesis, as well as the application of Vstat120 in the context of oncolytic virotherapy.
The manuscript is well written and structured. I only have two comments: a) the possibility of future studies on the action of BA1 depending on the types of angiogenesis, especially sprouting angiogenesis and its counterpart intussusceptive angiogenesis can be added, and b) although the references in the manuscript are adequate,  it is recommended to add some recent works on BAI, such as the review on BAI subfamily by Moon SY et al., Gene, 2018, and the contributions of Liu L et al., 2020 on BAI1 in lung cancer A549 cells and of Zhou M et al., 2021 on that the expression level of TARBP2 in human tumour tissue is negatively correlated with the expression of BAI1.

Author Response

Reviewer #4: In the manuscript entitled “Therapeutic Application of Brain-Specific Angiogenesis Inhibitor 1 for Cancer Therapy” by Mitra Nair, Chelsea Bolyard, Tae Jin Lee, Balveen Kaur *, Ji Young Yoo*, the authors review structure and function of the Brain-specific angiogenesis inhibitor 1 (BAI1/ADGRB1), an adhesion G protein-coupled receptor, and its application in the context of cancer therapy. They present the structure and functions of BAI1/ADGRB1 and summarize the works demonstrating that BAI1/ADGRB1 expression is inversely correlated with the tumour malignancy and survival prognosis in cancer. Likewise, they take into account other roles of BAI1/ADGRB1 on phagocytosis, myoblast fusion and synaptogenesis, as well as the application of Vstat120 in the context of oncolytic virotherapy.

The manuscript is well written and structured. I only have two comments:

  1. The possibility of future studies on the action of BA1 depending on the types of angiogenesis, especially sprouting angiogenesis and its counterpart intussusceptive angiogenesis can be added.

Response: We thank the reviewer for the suggestion. In the revised manuscript, we have added the future study on the action of BAI1 as below.

Although BAI1 is known as an anti-angiogenic factor, the detailed mechanisms of its anti-tumor and anti-angiogenic activity have yet to be elucidated. More importantly, in clinical trials, most conventional anti-angiogenic drugs target the sprouting angiogenesis, and unfortunately these drugs become resistant and have minimal clinical benefits [40]. Therefore, it would be interesting to study the mechanism of action of BAI1 depending on the type of angiogenesis, such as sprouting angiogenesis and intussusceptive angiogenesis.

  1. Although the references in the manuscript are adequate, it is recommended to add some recent works on BAI, such as the review on BAI subfamily by Moon SY et al., Gene, 2018, and the contributions of Liu L et al., 2020 on BAI1 in lung cancer A549 cells and of Zhou M et al., 2021 on that the expression level of TARBP2 in human tumour tissue is negatively correlated with the expression of BAI1.

Response: We thank the reviewer for the great suggestion. In the revised manuscript, we have added more recent works on BAI1.  

Round 2

Reviewer 1 Report

The article can be accepted in this form.

Reviewer 3 Report

The revised version of the manuscript still lacks of important connections between the sections. In this recent era we are full of data, hence a well-written and organized review should profoundly influence the readers.